# Genomic insights into the evolutionary relationships and demographic history of kiwi

**Michael V. Westbury**[1]*, **Binia De Cahsan**[1], **Lara D. Shepherd**[2], **Richard N. Holdaway**[3,4,5], **David A. Duchene**[1], **Eline D. Lorenzen**[1]*

**1** Globe Institute, University of Copenhagen, Copenhagen, Denmark, **2** Museum of New Zealand Te Papa Tongarewa, Wellington, New Zealand, **3** Palaecol Research Ltd, Christchurch, New Zealand, **4** School of Biological Sciences, University of Canterbury, Christchurch, New Zealand, **5** School of Earth and Environment, University of Canterbury, Christchurch, New Zealand

* m.westbury@sund.ku.dk (MVW); elinelorenzen@sund.ku.dk (EDL)

**Data Availability Statement:** All raw data used in this manuscript were previously published and the accession codes are listed in supplementary table S2. Example commands and the scripts necessary

## Abstract

Kiwi are a unique and emblematic group of birds endemic to New Zealand. Deep-time evolutionary relationships among the five extant kiwi species have been difficult to resolve, in part due to the absence of pre-Quaternary fossils to inform speciation events. Here, we utilise single representative nuclear genomes of all five extant kiwi species (great spotted kiwi, little spotted kiwi, Okarito brown kiwi, North Island brown kiwi, and southern brown kiwi) and investigate their evolutionary histories with phylogenomic, genetic diversity, and deep-time (past million years) demographic analyses. We uncover relatively low levels of gene-tree phylogenetic discordance across the genomes, suggesting clear distinction between species. However, we also find indications of post-divergence gene flow, concordant with recent reports of interspecific hybrids. The four species for which unbiased levels of genetic diversity could be calculated, due to the availability of reference assemblies (all species except the southern brown kiwi), show relatively low levels of genetic diversity, which we suggest reflects a combination of older environmental as well as more recent anthropogenic influence. In addition, we suggest hypotheses regarding the impact of known past environmental events, such as volcanic eruptions and glacial periods, on the similarities and differences observed in the demographic histories of the five kiwi species over the past million years.

## Introduction

New Zealand's unique and emblematic kiwi, comprising five extant species, represent a highly divergent avian lineage within Palaeognathae. Kiwi are the only members of the Apterygidae family, and are endemic to New Zealand. The species display a number of unusual biological attributes more commonly associated with small mammals. These include low metabolic rate, lack of colour vision, flightlessness, increased longevity, and nocturnality [1, 2].

The evolutionary origins of kiwi are yet to be fully elucidated. Kiwi are estimated to have diverged from their nearest known relative, the elephant birds (Aepyornithidae), ~50 million years ago (Ma) [3]. Elephant birds are only known from Madagascar, and it is therefore thought kiwi arrived in New Zealand via flight rather than Gondwanan continental movement

to perform all analyses in this manuscript are available at github.com/Mvwestbury/Kiwi-genomes.

**Funding:** This work was supported by the Independent Research Fund Denmark | Natural Sciences, Forskningsprojekt 1, grant no. 8021-00218B, and the Villum Fonden Young Investigator Programme, grant no. 13151, to EDL. The funders had no role in study design, data collection and analysis, decision to publish, or preparation of the manuscript.

**Competing interests:** The authors have declared that no competing interests exist.

and vicariance [4]. Despite such a deep divergence from its closest known relative, the oldest reported kiwi fossil has been dated to at least 19–16 Ma [5], and molecular data have suggested the root of living kiwi can be traced back to between ~15 Ma [6] and ~5 Ma [7]. Thus, wide uncertainty remains with regard to the lineage leading to extant kiwi, after Apterygidae diverged from Aepyornithidae. Furthermore, deep evolutionary relationships are obscured by the lack of pre-Quaternary fossils to inform speciation events among the extant species.

The five recognized species of kiwi all belong to the genus *Apteryx*. The species are placed into two morphologically and genetically distinct clades [7, 8] (Fig 1). One clade comprises two extant species, great spotted kiwi (*A. maxima*, formerly *A.haastii [9]*) and little spotted kiwi (*A. owenii*). The other clade includes Okarito brown kiwi (*A. rowi*, also known as rowi), southern brown kiwi (*A. australis*, also known as tokoeka), and North Island brown kiwi (*A. mantelli*). However, despite the deep divergences of these clades and lineages [3, 6, 7], interspecific hybridization between species from different clades has recently been reported [9].

Prior to the arrival of humans 1,000–800 years ago, kiwi were found from coastal to subalpine habitats across New Zealand, but were most common in lowland rainforest habitats [7]. Partly due to their unique traits, such as flightlessness and longevity, kiwi are highly vulnerable to mammalian predators. Since human colonisation, the distribution ranges of all five extant kiwi species have been greatly reduced (Fig 1). The conservation status of four of the five species is relatively critical (S1 Table in S1 File), and is attributed to population declines following human colonisation [10, 11].

Previous studies using short fragments of mitochondrial DNA retrieved from prehistoric and contemporary specimens revealed the genetic consequences of these past anthropogenically driven range contractions, and suggested significant loss of genetic diversity after the arrival of humans, especially in the Okarito brown, southern brown, and little spotted kiwi [11, 12]. In fact, all extant little spotted kiwi are believed to descend from five founding individuals that were placed on the predator-free Kapiti Island in 1912 [13].

The use of genome-wide data can greatly enhance our existing understanding of kiwi relationships, as they allow investigation of species' deep-time evolutionary histories. *De novo* nuclear genome assemblies are available from four of the five kiwi species (great spotted kiwi, little spotted kiwi, Okarito brown kiwi, North Island brown kiwi) [2, 14], and resequencing data are available from the southern brown kiwi [15]. Recent studies incorporating genomic data from kiwi have begun to address outstanding questions regarding demographic histories and divergences within kiwi. Palaeognaths have been reported to show large phylogenetic discordance across the genome, although the study did not focus on kiwi specifically [14]. Studies focussed on kiwi have reported monophyletic relationships among species, albeit with some signals of admixture between individual lineages [7, 15]. Furthermore, based on demographic history reconstructions of the past 100 thousand years it has been suggested changes in kiwi effective population sizes were likely driven by past changes in glacial extent [7, 15].

While these studies provide valuable insights into the evolutionary history of kiwi, an investigation of the presence of gene flow between lineages, and insights into deeper time (up to 1 Ma) demographic trends, is lacking. Here, we utilise single genome representatives of each extant kiwi species to better understand the interspecific evolutionary relationships among species, and to elucidate how past environmental changes may have impacted their demographic histories. Furthermore, due to the availability of reference genomes for four out of the five kiwi species, we evaluate the resilience of our phylogenomic and demographic results to reference genome biases.

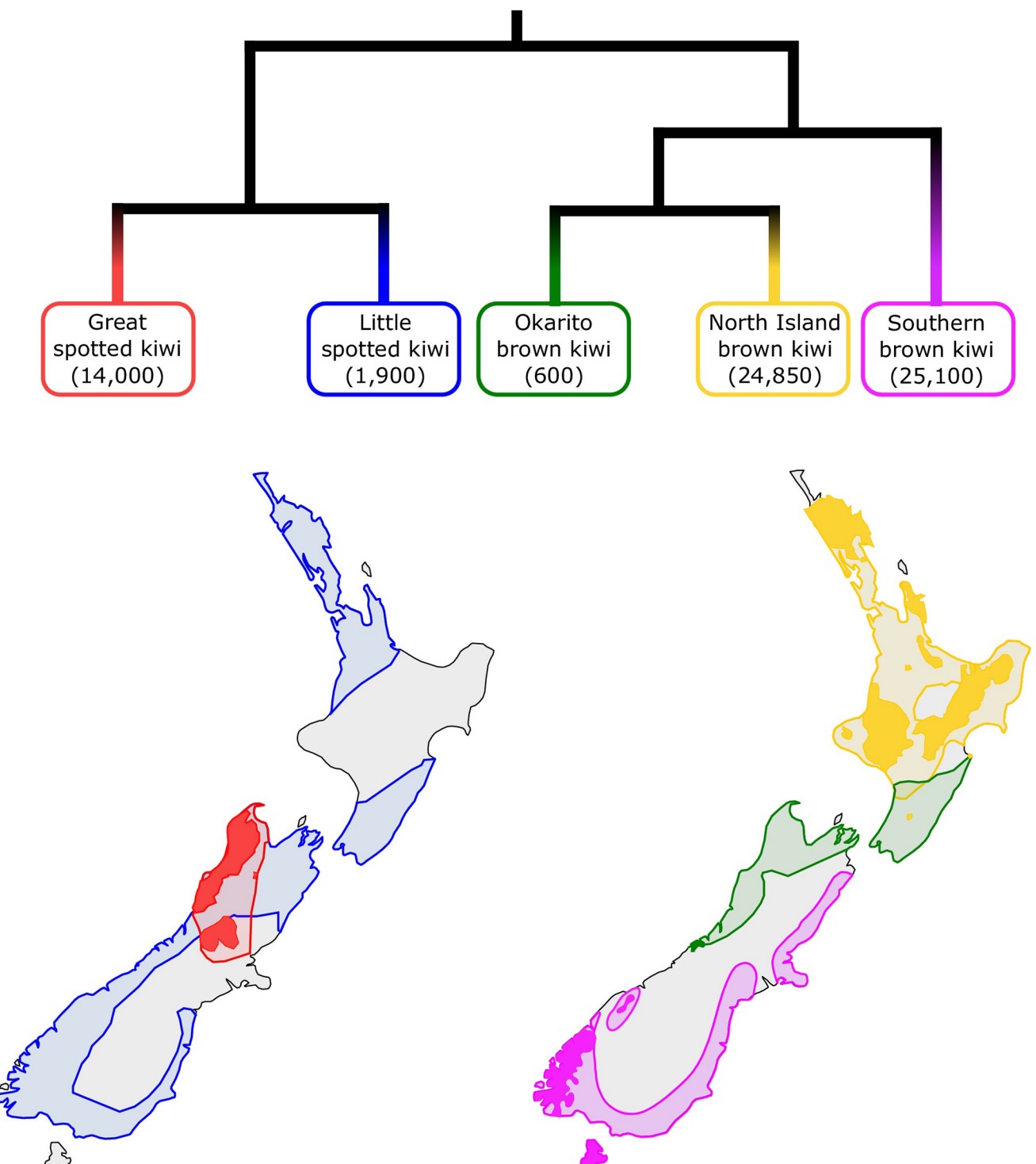

**Fig 1. Phylogenetic relationships and distribution ranges of the five extant kiwi species.** Modern and historic distributions are based on [51]. Lightly shaded areas show estimated distributions prior to human arrival and dark shading shows current distribution. Current distribution for the little spotted kiwi is not shown, as it is presently only found on small offshore islands and wildlife sanctuaries. Population size estimates are shown in parentheses under each species name [51].

## Materials and methods

### Data

We downloaded raw Illumina sequencing reads and genome assemblies from each of the four available kiwi species from Genbank: great spotted kiwi, little spotted kiwi, Okarito brown kiwi, and North Island brown kiwi [2, 14]. We downloaded the raw reads for a southern brown kiwi from the individual with the highest number of available reads [15]. To act as an outgroup for the phylogenomic analyses, we downloaded raw sequencing reads and the genome assembly of emu (*Dromaius novaehollandiae*) [14]. For estimates of comparative genetic diversity, we downloaded raw sequencing reads and the genome assemblies of three additional Paleognath species: ostrich (*Struthio camelus*) [16], southern cassowary (*Casuarius casuarius*) [14], and greater rhea (*Rhea americana*) [14]. Accession code details for all species can be found in S2 Table in S1 File and assembly statistics in S3 Table in S1 File.

### Data processing

We trimmed adapter sequences and removed reads shorter than 30 bp from the downloaded raw reads using skewer [17], mapped the trimmed reads to the specified reference genome assembly using BWA v0.7.15 [18] and the mem algorithm. We parsed the output and removed duplicates with SAMtools v1.6 [19]. Furthermore, as some downstream analyses required the removal of sex chromosomes, we independently determined which scaffolds were most likely autosomal in origin for each reference genome assembly used in the current study. We found putative sex chromosome scaffolds by aligning each reference genome assembly to the chicken (*Gallus gallus*) Z and W chromosomes (Genbank accessions: Z—CM000122.5, W—CM000121.5). Alignments were performed using satsuma synteny [20] and default parameters. Information on all mappings performed can be found in S4 Table in S1 File.

### Phylogenomics

We performed multiple phylogenomic analyses using different mapping reference genome assemblies, to assess the robustness of our results to reference choice. We independently mapped the raw reads of all five kiwi species to two reference genome assemblies: emu and great spotted kiwi. We also mapped the emu raw reads to the emu reference genome assembly. From the resultant mapped files, we built majority rules consensus sequences (-doFasta 2) in ANGSD v0.921 with the following parameters; -mininddepth 5 -minmapq 30 -minq 30 -uniqueonly 1, and only included autosomal scaffolds >100kb in length (-rf). We extracted 2 kb windows with a 1 Mb slide from each consensus sequence using bedtools v2.26.0 [21]. This gave us 1,546 windows when mapping to the emu and 1,889 windows when mapping to the great spotted kiwi. We filtered for windows that had at least 10 parsimony-informative sites and performed maximum likelihood phylogenetic inference for each remaining window in IQ-TREE2 [22], using the best GTR+R+F model according to the Bayesian information criterion [23]. Using the estimated gene trees (which in this case refers to a window), we inferred the species tree of kiwi under each reference genome assembly, using the summary multispecies coalescence as implemented in ASTRAL v4.10 [24]. The species tree inferred for each of the two datasets was used as the focal tree to calculate gene- and site-concordance factors for each branch in IQ-TREE2 [25].

### Quantifying introgression via branch lengths

To investigate whether phylogenetic discordance among all possible kiwi triplets [[A,B],C] can be explained by incomplete lineage sorting (ILS) alone, or by a combination of ILS and gene

flow, we implemented Quantifying Introgression via Branch Lengths (QuIBL) [26] on the dataset obtained when mapping the kiwi and emu raw read data to the emu reference genome. We used the kiwi and emu mapped to the emu dataset due to the requirement of an outgroup. Furthermore, since this analysis relies on the relative ages of nodes, we only used loci with nearly constant evolutionary rates among lineages. Specifically, we rooted each gene tree with the emu and excluded those loci with a coefficient of variation in root-to-tip length >0.01. We ran QuIBL specifying the emu as the overall outgroup (totaloutgroup), to test either ILS or ILS with gene flow (numdistributions 2), the number of total EM steps as 50 (numsteps), and a likelihood threshold of 0.01. We determined the significance of gene flow by comparing values of BIC1 (ILS alone) and BIC2 (ILS and gene flow). If the difference between BIC1 and BIC2 was greater than 10, as suggested by the original paper describing the method [26], we assumed incongruent topologies arose due to both ILS and gene flow. With a difference of less than 10, we assumed ILS alone.

### *f*-branch statistic

To further investigate the potential for gene flow between kiwi lineages, we implemented the *f*-branch test [27, 28] The test takes correlated allele sharing into account when visualising *f*4-ratio results meaning it can also uncover indications of gene flow between ancestral lineages. As input we created a multi-individual variant call file (VCF) using the data mapped to the emu genome and BCFtools v1.6 [29]. Specifically we used BCFtools mpileup, and filtered the VCF file to only include autosomal SNPs using BCFtools call and the -mv parameter. We ran the multi-individual VCF through Dtrios in Dsuite v0.4 r43 [27] and specified the species tree (Fig 1) and otherwise default parameters. We ran the output from Dtrios through *f*-branch and visualised the output using the dtools.py script from Dsuite. The default parameters for the *f*-branch statistic in Dsuite only consider *f*b, a value indicating excess allele sharing between a given branch (relative to its sister branch) and a non-sister branch, with p<0.01. However, we also assessed statistical significance of *f*b using a block Jack-knife approach by including the -Z parameter when running the *f*-branch statistic in Dsuite. We used the default number of 20 Jackknife blocks for this test. A Z score |Z|>3 was considered as significant.

### End of lineage sorting/gene flow

To estimate the point in time when the genomes of the five kiwi species had fully coalesced, putatively indicating an end of lineage sorting and/or gene flow, we used the F1 hybrid Pairwise Sequentially Markovian Coalescent model, hPSMC [30]. hPSMC utilises pseudo-diploid sequences by merging pseudo-haploid sequences from two different genomes, which in our case are from different species. As the variation in the interspecific pseudo-F1 hybrid genome cannot coalesce more recently than the emergence of reproductive isolation between the two parental species, we can use this method to infer when reproductive isolation between two species may have occurred. If some regions within the genomes of two target species are yet to fully diverge, due to ILS or to gene flow, hybridisation may still be possible.

Previous studies have shown that the results of hPSMC are not significantly influenced by mapping reference [31, 32], and we therefore performed each analysis once, with the raw reads from the five kiwi species mapped to the great spotted kiwi reference genome assembly. We constructed haploid consensus sequences for each of the kiwi individuals using the same consensus sequences as the phylogenomic analysis. We merged the resultant haploid consensus sequences pairwise into a pseudo-diploid sequence using a python script available as part of the hPSMC toolsuite. The resultant pseudo-diploid sequences were run through a Pairwise Sequentially Markovian Coalescent model (PSMC) [33]. We ran PSMC specifying standard

atomic intervals (4+25*2+4+6), a maximum number of iterations of 25 (-N), maximum 2N0 coalescent time of 15 (-t), and initial theta/rho ratio of 5 (-r).

To calibrate the PSMC plots, we calculated an *Apteryx* average mutation rate. We did this by first calculating the average pairwise distance for each species pair (S5 Table in S1 File), and dividing that by 2 * the previously published divergence times [3]. However, it should be noted that this calculation only provides an estimate and is influenced by divergence times. Given this, we selected the divergence times from Yonezawa et al [3] because they used the dataset with the largest number of genetic markers at the time and included all five kiwi species in their estimates and we therefore deemed it the most reliable. We calculated the pairwise distances in ANGSD v0.921 [34] using a consensus base call approach, with all species mapped to the great spotted kiwi reference genome assembly, and applying the same filters as for the phylogenomic analyses with the additions of only including sites found in all individuals (-minInd 5), and print a distance matrix (-makematrix 1). Given the possibility for ILS and gene flow in the kiwi lineages, whole genome pairwise distances may be under-estimated from whole genome alignments. This resulted in a mutation rate of $8.0 \times 10^{-10}$ per year or $2.0 \times 10^{-8}$ per generation, assuming a generation time of 25 years [7]. Our mutation rate of 2.0e-8 per generation is comparable to the rate of 1.34e-8 per generation from Bemmel et al [15].

From the PSMC output, we manually estimated the pre-divergence Ne of each pseudodiploid genome by outputting the text file (-R) using the plot script from the PSMC toolsuite. Using the pre-divergence Ne estimated from this output, we ran simulations to infer the intervals during which the pseudodiploid genomes coalesce between each species pair using ms [35]. Simulation commands in ms were automatically produced with the hPSMC_quantify_split_time.py python script from the hPSMC toolsuite, while specifying the pre-divergence Ne and the time windows we wanted to simulate, and the remaining parameters as default. The time intervals and pre-divergence Ne for each species pair can be found in S6 Table in S1 File. We plotted the results and found the simulations with an exponential increase in Ne to be closest to the empirical data, between 1.5-fold and 10-fold the value of the pre-divergence Ne. The divergence times from these simulations were taken as the time interval during which lineage sorting and/or gene flow stopped. We considered the portion between 1.5-fold and 10-fold the value of the pre-divergence Ne, as suggested in previous work [30]. This was done to capture the portion of the hPSMC plot most influenced by the divergence event. The lower bound is set to control for pre-divergence increases in population size, and the upper bound is to avoid exploring parameter space in which little information is present.

## Autosome-wide heterozygosity and inbreeding estimates

We calculated autosome-wide levels of heterozygosity and runs of homozygosity (ROH) for each species from the mapped bam files using the software ROHan [36]. The raw reads of the four species for which conspecific reference genome assemblies were available, were independently mapped to each said assembly. Raw reads of the southern brown kiwi, which does not have an available conspecific assembly, were mapped to the North Island brown kiwi assembly. We ran ROHan three times independently, using different window sizes (500 kb, 1 Mb, 2 Mb), while keeping the other parameters as default. The default parameters specify a window as being a ROH if it has an average heterozygosity of less than $1 \times 10^{-5}$. We calculated autosomewide heterozygosity for the non-kiwi palaeognath species (emu, ostrich, southern cassowary, greater rhea) in the same way, but using only a 1 Mb window size.

We estimated the number of generations since inbreeding occurred that each respective ROH length represents using $g = 100/(2rL)$ [37], where r = recombination rate, L = length of ROH in Mb, and g = number of generations. As genome-wide recombination rates for kiwi

are unavailable, we present results based on two, 3cM per Mb based on the chicken [38] and 2.1 cM/Mb based on the rhea [39] Given this calculation, ROH>0.5Mb equates to inbreeding occurring within the last 47.6 or 33.3 generations, ROH>1Mb equates to inbreeding occurring within the last 23.8 or 16.6 generations, and ROH>2Mb equates to inbreeding occurring within the last 11.9 or 8.3 generations.

### Demographic reconstruction

We ran demographic analysis on diploid consensus genomes from each kiwi species, each mapped to their conspecific reference genome assembly and the southern brown kiwi mapped to the North Island brown kiwi, using PSMC [33]. We called diploid genome sequences using SAMtools and BCFtools, specifying a minimum quality score of 20 and minimum coverage of 10. However, we also created a diploid consensus genome with Okarito brown kiwi mapped to the North Island brown kiwi, as it has previously been shown that mapping to the Okarito brown kiwi reference genome assembly may be problematic for PSMC analysis [40]. As Prasad et al. [40] did not assess the reliability of the little spotted kiwi as mapping reference, we further assessed the reliability of the little spotted kiwi PSMC results by mapping to the conspecific reference genome assembly, as well as to the great spotted kiwi and North Island brown kiwi reference genome assemblies. Before running PSMC, we removed scaffolds found to align to sex chromosomes in the previous step, and removed scaffolds shorter than 100 kb. We ran PSMC specifying the same parameters as in the hPSMC analysis and performed 100 bootstrap replicates to investigate support for the resultant demography. PSMC results were checked for overfitting so that after 20 rounds of iterations, at least 10 recombinations are inferred to have occurred in the intervals each parameter spans. We plotted the output using a mutation rate of $2.0x10^{-8}$ per generation, assuming a generation time of 25 years as detailed above for the hPSMC analysis.

## Results

### Mapping results

The number of reads mapping as well as the average genome-wide coverages slightly changed depending on whether individuals were mapped to a con- or heterospecific reference genome, with lower values being recovered when mapping to a heterospecific reference (S4 Table in S1 File). When mapping to conspecific reference genomes, we recovered genome-wide coverages of 24.52x for the great spotted kiwi, 20.8x for the North Island brown kiwi, 31.6x for the little spotted kiwi, and 25.2x for the Okarito brown kiwi. Mapping the southern brown kiwi to the North Island brown kiwi reference genome gave 14.8x.

### Interspecific evolutionary relationships

Reads more similar to the reference map more successfully than divergent reads, artificially inflating signals of genetic similarities between a highly divergent outgroup and an ingroup species used as mapping reference [41]. As our outgroup (emu) is highly divergent from the ingroup kiwi species, our results when mapping to the ingroup great spotted kiwi could therefore be biased by the emu artificially appearing more genetically similar to the great spotted kiwi than the non-reference species. Although fewer reads mapped successfully (S4 Table in S1 File), by also mapping to the outgroup we tested for the influence of this potential bias. Our results are unlikely to be driven by reference bias as, regardless of the genome assembly selected as mapping reference, the vast majority of windows in our phylogenomic analysis support the grouping of little spotted kiwi and great spotted kiwi as sister species, and the

grouping of Okarito brown kiwi, North Island brown kiwi, and southern brown kiwi in a separate clade, with the southern brown kiwi being sister to the Okarito brown kiwi, North Island brown clade (Fig 1, S7 Table in S1 File). Despite the overall support for the species tree, within brown kiwi, alternative gene tree topologies occurred at relatively high frequency compared to other alternative topologies found (up to 34.9%). Site concordance factors also found high levels of discordance within this clade (up to 42.1%).

Further examination using QuIBL to assess if alternative topologies may have arisen due to ILS alone, or to both ILS and gene flow, indicated a mixture of both ILS and gene flow causing the topological discordances in our dataset (S8 Table in S1 File). QuIBL suggests gene flow between great spotted kiwi and the Okarito brown kiwi, great spotted kiwi and southern brown kiwi, North Island brown kiwi and southern brown kiwi, Okarito brown kiwi and little spotted kiwi, and southern brown kiwi and little spotted kiwi.

The $f$-branch statistic suggests several signals of gene flow between kiwi lineages, as shown by elevated $f_b$. We find elevated $f_b$ between i) the little spotted kiwi and all three brown kiwi species, ii) the southern brown kiwi and both the great and lesser spotted kiwi, and iii) the Okarito brown kiwi with both spotted kiwi and the southern brown kiwi (Fig 2). All results were significant with Z>3 (S9 Table in S1 File).

To quantify when reproductive isolation may have been complete between kiwi lineages, we ran the F1 hPSMC described above. Between the two major clades, we found that lineage sorting and/or gene flow ceased 2.6–1.8 Ma. This pattern was the same, regardless of which individuals were used in the pairwise comparison (S1 Fig in S1 File). Within clades, we find complete reproductive isolation (lineage sorting was complete and/or gene flow ceased) between great spotted kiwi and little spotted kiwi 700–400 thousand years ago (kya) (S2 Fig in S1 File), and similarly between North Island brown kiwi and Okarito brown kiwi 800–500 kya (S3 Fig in S1 File). Lineage sorting was complete and/or gene flow ceased 1.2 Ma-800 kya between southern brown kiwi and both the North Island brown kiwi and the Okarito brown kiwi (S4 Fig in S1 File). hPSMC uses the PSMC method, and therefore changes in effective population size can be influenced by population structure [42] and regions undergoing strong negative selection [43]. However, in our hPSMC when comparing different species pairs, e.g. southern brown kiwi/North Island brown kiwi and southern brown kiwi/Okarito brown kiwi, we get similar if not identical results. Therefore, we do not think population structure largely influences hPSMC results. Selection on the other hand could play a role and remains to be tested. However, as previous studies showed strongly deleterious mutations mask declines in PSMC [43], and we are focusing on exponential increases, we do not think our results will be greatly influenced by this.

## Autosome-wide heterozygosity and inbreeding estimates

We find the highest levels of autosome-wide heterozygosity in North Island brown kiwi, followed by great spotted kiwi, southern brown kiwi, Okarito brown kiwi, and little spotted kiwi (Fig 3A). To further contextualise these estimates, we calculated the autosome-wide heterozygosity estimates of four other palaeognaths (emu, ostrich, southern cassowary, greater rhea). Our analysis revealed that kiwi species have relatively low levels of heterozygosity compared to the other palaeognaths, except for the southern cassowary, whose heterozygosity only exceeded that of little spotted kiwi.

We further investigated whether recent inbreeding may have caused the low levels of autosome-wide heterozygosity using runs of homozygosity (ROH) analyses. As expected, the number of ROH increased when specifying increasingly smaller window sizes as a ROH (Fig 3B). Moreover, the general trend reflected what would be expected based on autosome-wide

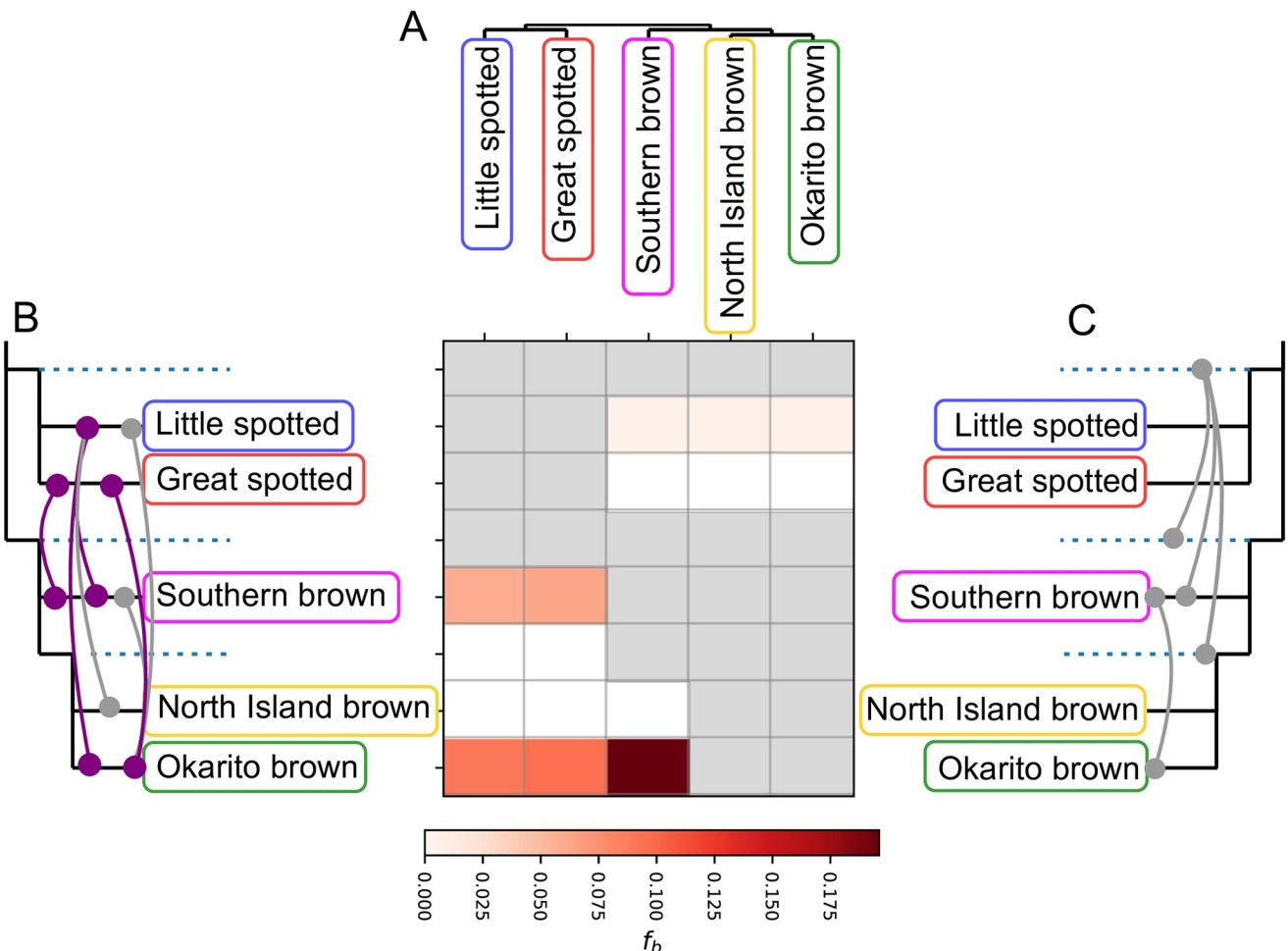

**Fig 2. Genome-wide *f*-branch results.** (A) Species tree; (B) and (C) Species tree in expanded form, with internal branches as dotted lines. The values in the matrix refer to excess allele sharing between the expanded tree branch (relative to its sister branch) and the species on the *x*-axis. Grey squares are comparisons that could not be made due to the topological input requirements of the test and its inability to infer gene flow between sister lineages. Lines connecting branches show: (B) gene flow events inferred directly from the *f*-branch results. Dark purple coloured lines show results supported by QuIBL; (C) gene flow events that we hypothesise from the *f*-branch results, while accounting for (i) the inability to detect gene flow between sister lineages, and (ii) a lack of a positive means less gene flow relative to the sister lineage, rather than no gene flow.

heterozygosity levels. That is, the species with the lowest overall heterozygosity (little spotted kiwi) also had the most ROH, and the species with the highest overall heterozygosity (North Island brown kiwi) had the least ROH. We found the lowest levels of ROH in the southern brown kiwi (Fig 3B).

This pattern was not as obvious in the other palaeognath species. The individual with the highest levels of diversity (rhea) also showed considerable levels of ROH (1 Mb window size; 2.2% of the genome in ROH). However, the individual with the lowest diversity (southern cassowary), also had the highest levels of ROH (1 Mb window size; 8.42% of the genome in ROH). The remaining two species had low levels of ROH: emu—0.50% and ostrich—0.64%.

## Intraspecific demographic histories

Our assessment of how mapping reference influences PSMC results of little spotted kiwi showed similar results to those reported previously for Okarito brown kiwi [40]; we find

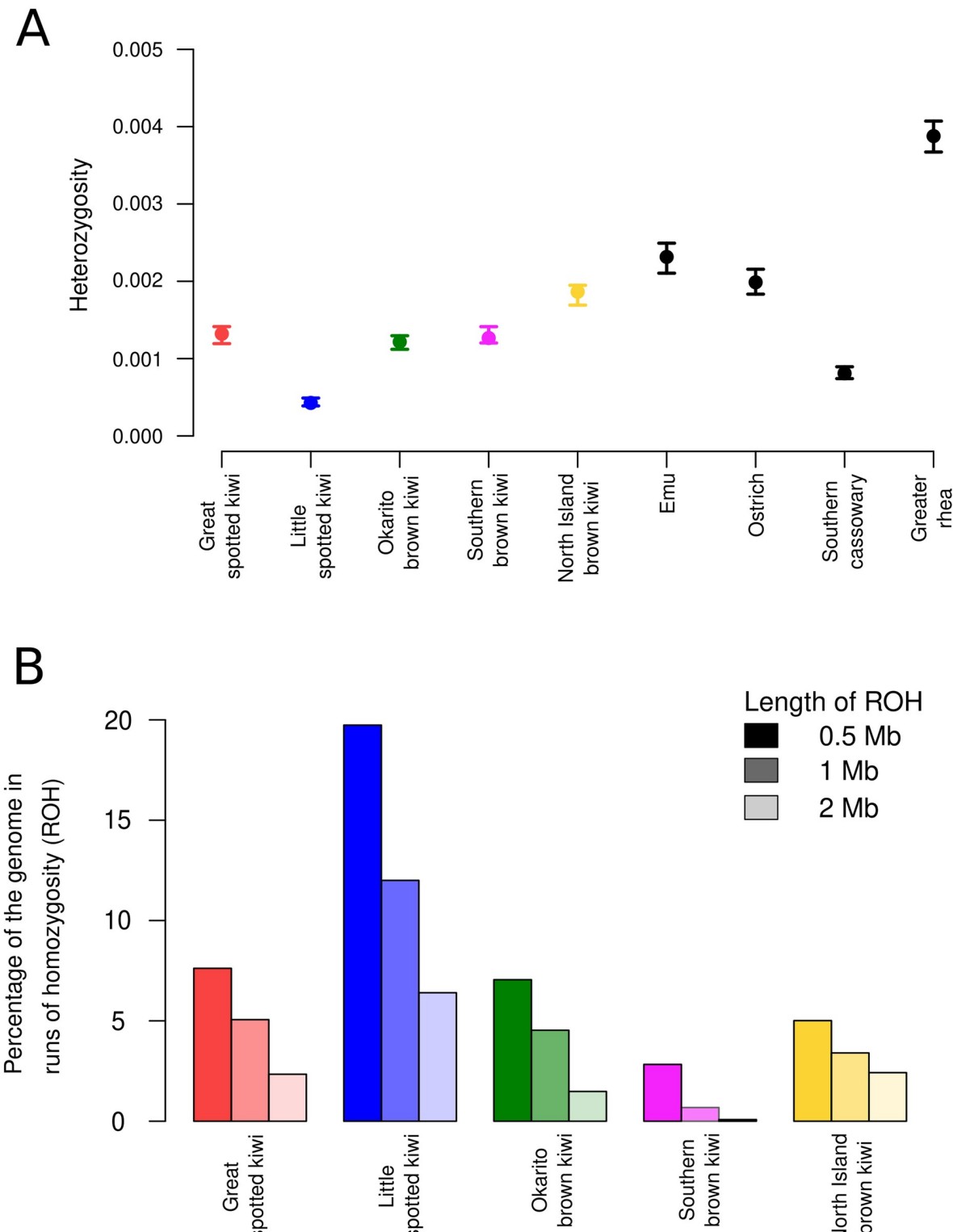

**Fig 3. Genetic diversity and inbreeding estimates.** (A) Autosome-wide heterozygosity calculated using ROHan for the four kiwi species with conspecific reference genomes, and four other palaeognath species. Error bars show maximum and minimum values. (B) Autosomal runs of homozygosity calculated using various window sizes (0.5 Mb, 1 Mb, and 2 Mb) for the four kiwi species with conspecific reference genomes. Differential shading shows the window size used.

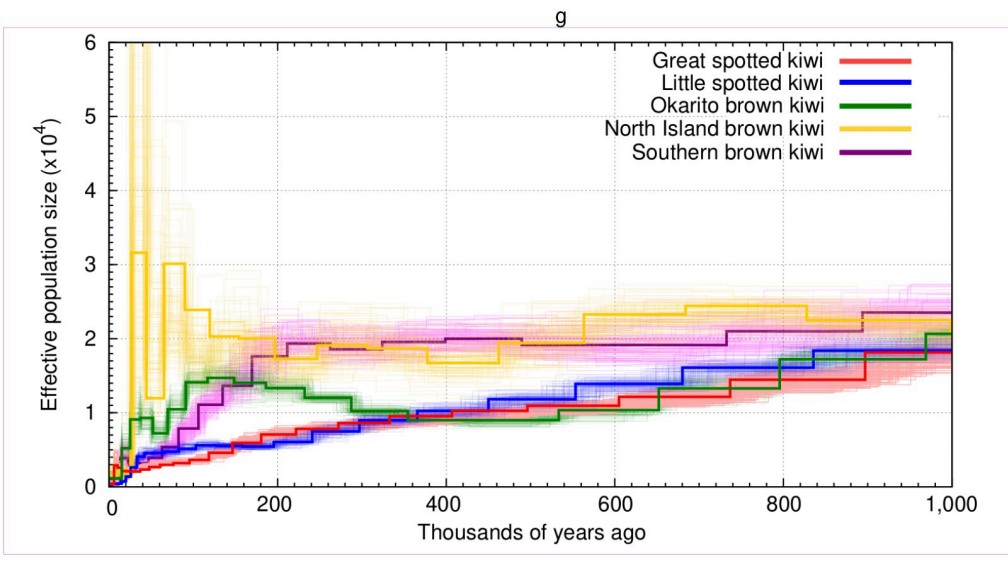

**Fig 4. Effective population sizes over the past one million years calculated using PSMC.** Faded lines show the 100 bootstrap replicates produced for each species.

relatively consistent results when mapping the little spotted kiwi raw reads to both North Island brown kiwi and to great spotted kiwi, but much reduced effective population size (Ne) values when mapping to little spotted kiwi (S5 Fig in S1 File). Therefore, we based our inferences on the results retrieved when mapping to its closest relative (great spotted kiwi).

We find Ne in North Island brown kiwi was relatively high and stable over the past 1 Ma, with several fluctuations within the past 100–50 kya (Fig 4). However, the bootstrap support values during the latter time period show considerable uncertainty, suggesting little confidence in these fluctuations. The southern brown kiwi also exhibited relatively stable Ne values until ~200 kya where there was a continual decrease in Ne until present. Ne in Okarito brown kiwi decreased 1–0.5 Ma, followed by a plateau and slight increase, then a more rapid decline in the past ~100 kya. The two remaining species, little spotted kiwi and great spotted kiwi, exhibit similar demographic trends. Both species show a continuous gradual decline in Ne over the past 1 Ma.

## Discussion

### Phylogenetic relationships

Based on our phylogenetic analysis, we found relationships among the five kiwi species to be generally conserved across windows, consistent with previous findings based on mtDNA sequences and nuclear data [3, 7, 15, 44] (Fig 1, S7 Table in S1 File). However, despite the clear overall species tree, gene (window) tree and site concordance factors both show the presence of discordant topologies across the genome, which QuIBL suggests are due to a mixture of ILS and gene flow. All signals of gene flow found in QuIBL were supported by our $f$-branch statistic results, to the exclusion of gene flow between the North Island and southern brown kiwis, which was not found in the $f$-branch statistic (Fig 2B). Furthermore, our $f$-branch statistic found several signals of gene flow not found with QuIBL. However, interpretations of the $f$-branch statistic can be complicated by results only being relative to a sister species, i.e. an $f_b$ of 0 does not mean no gene flow, but just less gene flow than the sister species. Taking this into account and the known order of divergence of each lineage [3], we hypothesise that some

observed signals of gene flow may be due to retention of introgressed loci from ancestral gene flow events (Fig 2C). Little spotted kiwi showed similar $f_b$ with all three brown kiwi. This pattern could reflect independent gene flow events between little spotted kiwi and each brown kiwi species (Fig 2B), or gene flow between little spotted kiwi and the ancestral brown kiwi. Little spotted and great spotted kiwi split after southern brown split from North Island and Okarito brown kiwi, and hence little spotted kiwi alone could not have exchanged genetic material with the ancestral brown kiwi. Rather, the ancestral spotted kiwi could have exchanged genetic material with the ancestral brown kiwi, and little spotted has retained more signal of this relative to great spotted kiwi (Fig 2C). Similarly, the signal between southern brown kiwi and both the little and great spotted kiwi, as well as between the Okarito brown kiwi and both the little and great spotted kiwi, could also reflect gene flow between ancestral lineages, with the retention of less of the introgressed alleles in the North Island brown kiwi (Fig 2C). Our hPSMC results (S1-S3 Figs in S1 File) suggest coalescence between species pairs occurred relatively recently, which can be interpreted as gene flow among all species after they initially diverged. A scenario of interspecific gene flow is congruent with recent reports of hybridisation between kiwi sister species, and between kiwi species from each of the two distinct clades [9].

The presence of interspecific gene flow between kiwi species may explain the large discrepancies between molecular estimates of their divergence times. However, varying estimates could also reflect difficulties in time calibration given uncertainties in fossil dates, substitution rate, and generation time. Estimates of the stem divergence of the five extant species have differed markedly in the literature. These include, but are not limited to, ~14.5 Ma [6], ~13.4 Ma [45], ~12.31 Ma [3], ~11.3 Ma [44], ~8.5 Ma [4, 46], and ~5.9 Ma [7]. Even when considering the youngest divergence estimate (~5.9 Ma), our hPSMC results suggest the genomes completely coalesced between 2.6 Ma and 1.8 Ma, which may be interpreted as the continuation of lineage sorting for a long time post divergence, or continued gene flow.

Interspecific gene flow is well documented in avian species [47] and may be due to the relatively slow rate at which postzygotic incompatibilities accumulate in birds [48]. Therefore, premating isolation is likely an important mechanism for the maintenance of reproductive isolation in birds, although this may be hampered by high dispersal rates through flight. Kiwi are known to have evolved flightlessness independently from other palaeognaths after their arrival to New Zealand [3, 6]. However, a lack of fossils makes it difficult to reliably determine when flightlessness evolved. Nevertheless, based on a recent fossil find, kiwi are thought to be flightless from at least the Middle Pleistocene (~1 Ma) [49]. Although a lack of migration through flight likely would have limited the dispersal capabilities of kiwi, some species had partly overlapping ranges in historic times (Fig 1), which may have facilitated hybridisation.

## Current and recent demography

Our finding of very low diversity in little spotted kiwi is congruent with previous studies based on microsatellite data [50] and population-level nuclear genomes [15]. Our finding—at least as represented by our sampled individual—may reflect high levels of recent inbreeding. Its high proportion of ROH >2Mb equates to inbreeding within the last ~250 years, and is congruent with current knowledge of the species. In 1912, five little spotted kiwi were moved to an offshore, predator-free island (Kapiti Island). Therefore, the entire population of the species, estimated at approximately 2,000 individuals [51] (Fig 1), is descended from at most five founders; a recent microsatellite study indicated perhaps only three founder individuals [12]. Such a low number of founders would have led to high levels of inbreeding. However, this findings contrasts with no inbreeding reported in a recent population genomic study [15]. Mapping to a phylogenetically distant reference can inflate heterozygosity estimates and

remove signs of inbreeding [40]. The finding of no inbreeding on a genome-wide level [15] may be a byproduct of the computational approach employed, rather than a biological signal; the little spotted kiwi data were not mapped to a conspecific reference genome and hence the lack of inbreeding may be the result of mapping biases.

Okarito brown kiwi experienced a bottleneck of ~150 birds in the 1990s [52]. However, we find similar diversity and inbreeding levels to great spotted kiwi (Fig 3) which has an estimated population size of 14,000 [51] (Fig 1). Furthermore, based on 11 microsatellite loci, it was reported that great spotted kiwi had high levels of genetic diversity compared with other kiwi species, and showed no evidence of a recent bottleneck (within the last 100 generations) [53]. Therefore, if genetic diversity alone predicts population size, great spotted kiwi should display higher levels of diversity than Okarito brown kiwi. However, as our Okarito brown kiwi individual was sampled in 1993, our findings likely do not reflect the genetic impact of the recent bottleneck in the 1990s. Therefore, the Okarito brown kiwi population at Okarito, from which this sample arose, likely had higher levels of diversity pre-bottleneck, and exemplifies the relevance of sample age when investigating recent bottleneck events. Finally, our finding of North Island brown kiwi having the highest levels of diversity is consistent with higher census population size (25,000) than the other species [51], and is congruent with a recent population genomic study [15].

## Long-term demography

Owing to a lack of fossil evidence, it is difficult to investigate how kiwi populations responded to past climatic and environmental events. Genomic data allows the exploration of changes in population size over evolutionary timescales to provide hypotheses based on correlations with known environmental events. Previous work using kiwi genomic data to investigate changes in effective population size have been interpreted within the framework of glacial and interglacial periods [7, 15].

We observe relatively stable Ne in the North Island brown kiwi over the past one million years (Fig 4). During this period, the central North Island experienced several major rhyolitic eruptions, as evidenced by tephras in ocean cores to the east and northeast of the North Island [54]. The ash from these eruptions rarely reached the northern peninsula of the North Island, providing a putative refuge for this population. Therefore, the northern distribution of North Island brown kiwi (Fig 1) may have facilitated its long-term stability in effective population size. Similarly, the southern range of the southern brown kiwi may have facilitated a stable Ne in this species 1 Ma—100 kya, despite eruptions.

In contrast, both great and little spotted kiwi show a general trend of declining Ne throughout the past one million years (Fig 4). As the genetic data of both species are from the South Island, despite the little spotted kiwi also occurring in the North Island (Fig 1), these results may only represent changes to South Island populations. Both volcanic fallout and repeated glaciations–or repeated *de*glaciations–may have had long-term negative effects on both kiwi species and caused decreases in census population size and/or connectivity, which would lead to the observed decrease in Ne.

The Ne of Okarito brown kiwi also shows a decreasing trend, suggestive of similar mechanisms to those in the great and little spotted kiwi (Fig 4). However, the trajectory deviates with a plateau in Ne, before rapidly decreasing ~100 kya. Associated with this decline is the end of the last interglacial, Isotope Stage 5e (130–116 kya [55]). A change in Ne at this time could be caused by the restriction of population movement, changes in population structuring, and/or a reduction in population size, all of which could have been driven by the environmental instability of this period. However, Okarito brown kiwi has experienced many glacial and interglacial periods before this event, and therefore it is difficult to determine correlation or causation.

Although the confidence in PSMC for more recent periods of time (<20kya) is limited [33], it is notable that the decline in Ne observed across all five kiwi species within the last 40 kya broadly coincides with the most recent global volcanic super eruption (Fig 4). A decrease in Ne could be due to a decrease in absolute population size, or a decrease in connectivity between previously connected populations. Similar declines in the past 10–20 kya have previously been reported [15]. The Oruanui eruption of Taupo volcano 25.6 kya deposited 1000 km$^3$ of tephra across New Zealand, from Auckland in the North Island to the Waitaki River in the South Island [56, 57]. Such a catastrophic event is expected to have destroyed most of the New Zealand biota between 38˚ and 45˚ south. If the decline in Ne in association with this major eruption was exhibited by just one taxon, little case could be made for cause and effect. However, the analogous temporal pattern of a population decline across all five species provides strong support for a significant environmental driver.

It is difficult to precisely pinpoint the driver of past population fluctuations, especially given the uncertainty in mutation rates and generation times. However, we form a number of hypotheses mostly based on correlations with the timing of volcanic activity. To date, the few studies using genomic data to investigate demographic histories of kiwi have alluded to glaciation as a major influence [7, 15]; although the Taupo caldera volcano ~1800 years ago has been identified as having some impact on kiwi Ne [15, 58]. By including more ancient time periods (>20kya) that experienced multiple volcanic events, we suggest volcanic eruptions may have been a previously overlooked major evolutionary force shaping the biota of New Zealand.

## Supporting information

**S1 File. File containing all supplementary information: S1-S9 Tables and S1-S5 Figs.**
(DOCX)

## Acknowledgments

We would like to acknowledge the Te Parawhau Trust and Waikaremoana iwi, Te Rūnanga o Ngāi Tahu, Te Ātiawa Manawhenua Ki Te Tau Ihu Trust, who provided guidance to the authors who generated the kiwi genomic data, on which our study is based. We would also like to thank the reviewers for their comments on improving the manuscript. The authors declare no conflicts of interest.

## Author Contributions

**Conceptualization:** Michael V. Westbury.

**Formal analysis:** Michael V. Westbury, Binia De Cahsan, David A. Duchene.

**Funding acquisition:** Eline D. Lorenzen.

**Supervision:** Michael V. Westbury, Eline D. Lorenzen.

**Writing – original draft:** Michael V. Westbury.

**Writing – review & editing:** Michael V. Westbury, Lara D. Shepherd, Richard N. Holdaway, Eline D. Lorenzen.

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
