## [Decision Letter · Decision Letter 0]

29 Apr 2022

PONE-D-22-08093Genomic insights into the evolutionary relationships and demographic history of kiwiPLOS ONE

Dear Dr. Westbury,

Thank you for submitting your manuscript to PLOS ONE. After careful consideration, we feel that it has merit but does not fully meet PLOS ONE’s publication criteria as it currently stands. Therefore, we invite you to submit a revised version of the manuscript that addresses the points raised during the review process (detailed below).

We look forward to receiving your revised manuscript.

Kind regards,

Susanne P. Pfeifer

Academic Editor

PLOS ONE

Journal Requirements:

"This work was supported by the Independent Research Fund Denmark | Natural Sciences, Forskningsprojekt 1, grant no. 8021-00218B, and the Villum Fonden Young Investigator Programme, grant no. 13151, to EDL. We would like to acknowledge the Te Parawhau Trust and Waikaremoana iwi, Te Rūnanga o Ngāi Tahu, Te Ātiawa Manawhenua Ki Te Tau Ihu Trust, who provided guidance to the authors who generated the kiwi genomic data, on which our study is based. The authors declare no conflicts of interest."

"This work was supported by the Independent Research Fund Denmark | Natural Sciences, Forskningsprojekt 1, grant no. 8021-00218B, and the Villum Fonden Young Investigator Programme, grant no. 13151, to EDL. The funders had no role in study design, data collection and analysis, decision to publish, or preparation of the manuscript."

Additional Editor Comments:

I have now received reviews from two experts in the field and their reading matches my own. Namely, given that this manuscript is based on publicly available genomic data, the authors will need to put their work in the appropriate context to demonstrate their novel contributions beyond the previously published work. Each reviewer makes additional points of importance with regards to the methodology as well as the interpretation of the results that would also need to be fully addressed. Of particular importance, the authors will need to provide evidence that the inferred model actually matches their data (and that PSMC is able to recapitulate the observations made from the data simulated under the inferred model). In this regard, the authors need to be mindful about the fact that their dataset contains regions affected by direct and linked selection, which can result in serious mis-inference of the population's demographic history (see Johri et al. MBE 2021). To allow the authors to address these comments / concerns, I recommend a major revision.

Reviewers' comments:

Reviewer's Responses to Questions

**Comments to the Author**

1. Is the manuscript technically sound, and do the data support the conclusions?

Reviewer #1: Partly

Reviewer #2: Partly

2. Has the statistical analysis been performed appropriately and rigorously? 

Reviewer #1: No

Reviewer #2: Yes

3. Have the authors made all data underlying the findings in their manuscript fully available?

Reviewer #1: Yes

Reviewer #2: Yes

4. Is the manuscript presented in an intelligible fashion and written in standard English?

Reviewer #1: Yes

Reviewer #2: Yes

5. Review Comments to the Author

Reviewer #1: This project uses previously published genomic data from five species of kiwi birds (and their close relatives, as outgroups) to understand the evolutionary history of this species. The project uses a set of population genomic methods to infer divergence times, introgression history, and inbreeding.

This manuscript has many strengths.

1. I like that the manuscript has a tight focus. While I always enjoy reading a paper that has broad implications, it is also always really nice to read a paper that focuses on the system and the data. It is nicely concise.

2. The study very carefully handles reference bias, which few comparative genomic projects even consider.

3. The figures are generally clear and easy to follow.

I have a few major suggestions for improvement.

1. There have been two really comprehensive papers published on kiwi evolutionary dynamics already, both out of Jason Weir's research group. This study cites both papers and references them a few times, but I found more should be done here to (1) explain in the introduction how this study can add to these existing studies and (2) compare results found across studies in the Discussion. Especially because the discussion is so focused on the kiwi clade itself, it should more strongly integrate what we already knew about kiwi diversification.

2. The study uses hPSMC (as I understand it) to infer gene flow between the species. I have never used hPSMC, but I have used PSMC, so I was curious about this application. I read the manual, and I cannot find any evidence that hPSMC allows inference of gene flow. I would recommend the authors either (1) clarify how they modified hPSMC to allow inference of gene flow or (2) clarify the language so that it no longer seems like hPSMC was used for gene flow inference. The discrepancies between divergence times inferred from a phylogeny could be due to ILS, post-divergence gene flow and / or methodological uncertainty. Given the QuIBL results, ILS and / or methodological uncertainty seems most likely.

3. The study often refers to the importance of ensuring that the reference genome used doesn't introduce bias. Given that this is a recurrent theme in the manuscript, I would suggest discussing this aspect of the study in the introduction and / or discussion to summarize what is known about this issue and why it matters.

MINOR COMMENTS

- L19 - 20: I might be over-interpreting this sentence, but fossils are unlikely to help infer evolutionary relationships -- just evolutionary timings. (See again L61 - 63.)

- L58 - 60: "In contrast" -- what are the authors contrasting here?

- L82: the Bemmels et al 2021 study also showed this, using whole genome data.

- L101 - 104: Recommend listing species names here again.

- L135: Consider reporting the total number of gene trees inferred.

- L154 - 156: What is the justification for these BIC thresholds?

- L159 - 169: Consider expanding the discussion of the hPSMC approach here. Most readers will be familiar with the PSMC approach, but might not fully appreciate the difference between PSMC and hPSMC.

- L171: What is inferred here is technically the substitution rate, no?

- L199: For this analysis, which variant set was used? The one inferred using ANGSD (L130)?

- L236: Recommend striking "earliest to diverge" here; see this great blog post here for why: http://for-the-love-of-trees.blogspot.com/2016/09/the-ancestors-are-not-among-us.html

- L237 - 238: 11% and 26% actually not that high! I would recommend reviewing some other phylogenomic discordance papers.

- L241: To my interpretation, the QuIBL results very strongly indicated this is likely just ILS.

- L243 - 253: I find this analysis a bit confusing. The study is comparing divergence times inferred from the phylogeny (which, it would be good to remind the readers here of how these were inferred and how deep they were) to timings inferred from hPSMC? We would expect these two estimates to be very different because they are measuring very different aspects of divergence history. This paragraph makes it clear that this discrepancy could be because of ILS or post-divergence gene flow. Yet, the first sentence of this paragraph seems to suggest this analysis was done to tell us about post-divergence gene flow, but this approach isn't the best approach to study post-divergence gene flow. (Also, another possibility for why these estimates are off: the difficulties in time calibration given uncertainties in substitution rate, fossil dates, and generation time. See also L318 - 320 for the same.) The study appears to be more conclusive that it is post-divergence gene flow in L431, which confuses me.

- L273 - 278: I would consider striking data on the other palaeognath species as they aren't really part of the paper otherwise.

- L304 - 306: Can the authors clarify how the hPSMC results show that there was continual gene flow? The high levels of discordance at the site level (Table S6) are in-line with what has been seen in other phylogenomic studies and could simply be the result of ILS.

- L322 - L323: High dispersal in birds is one hypothesis for why there is more hybridization, but another hypothesis is that birds seem to evolve postzygotic isolation slowly (Price an Bouvier 2002)

- L327 - L329: One aspect that confuses me here - these species seem to have abutting / overlapping ranges in the present day. Also current ranges do not necessarily reflect historical ranges. So, it is quite possible these species were co-occurring, which would have allowed hybridization pretty easily. And, even with no flight, these birds still disperse? Or is dispersal essentially non-existent?

- L361 - 3: Here is an example where the Bemmels paper should definitely be cited and used as a comparison point, as they conducted a more formal analysis of the same question.

- L422 - 428: Here is another place where the previous demographic work on kiwi should be cited.

- Figure 1: What is the citation for the historical ranges for these species?

Reviewer #2: In this study, Westbury et al. analyze representative whole genomes from each of five kiwi species to investigate the evolutionary history of this group. They primarily assessed phylogenetic relationships and looked for evidence of phylogenetic discordance due to gene flow and/or ILS, and they inferred historical effective population sizes with PSMC. The authors recover a phylogeny consistent with prior studies, and find some evidence of gene flow and/or ILS, although these latter results are not well resolved. Further, the authors find overall levels of genome-wide diversity that appear consistent with the effective population size histories (Ne of ~25k or less over the past million years), and some ROH, potentially indicative of inbreeding, in the little spotted kiwi.

The study exclusively uses data generated from other studies, which also evaluated the evolution and origins of kiwi. However, these prior studies are not sufficiently acknowledged in this manuscript. The data are not tied to their original sources (Sackton et al. 2019, Weir et al. 2016, Bemmels et al. 2021, more?), and the major findings and unresolved issues from the prior studies are not provided. Further, some important details from the methods are missing, and some of the results appear not to support one another (i.e. the ILS and gene flow analyses). Lastly, some of the conclusions based on the Ne trajectories relating to environmental shifts are unsupported. Given the work that has been done before, I am not sure what the contribution of this study is. In terms of its scope and aims, it is not well differentiated from the previous studies, which are probably more suited to investigating the evolutionary history of this system because they included more data. Overall, the manuscript was relatively clear and the analyses are mostly appropriate, but there are a number of crucial areas that require improvement. What follows is a detailed breakdown of my concerns.

89: The summary of prior genomic studies of kiwi in this introduction is insufficient. As written, the introduction fails to acknowledge that there are a few genomic studies of kiwi already in existence, which also tackle the issues of species origin, population structure, diversity, phylogenetic discordance, etc. with far more samples (e.g. Sackton et al. 2019, Weir et al. 2016, Bemmels et al. 2021). The major findings of those studies and the remaining gaps in knowledge should be presented in the introduction. How the present study addresses those gaps should be stated. How the results from this study agree with, add to, or refute prior studies must also be presented in the Discussion.

101: Citations are needed for all sequence data and reference genomes used in this study. This information should be added to the methods here and also included in Table S1. Also, in Table S1 the accession codes for the little spotted kiwi are duplicated - typo?

115: What is the depth of sequencing coverage? This information is essential for assessing the robustness of the methods and results.

132: How many sequences were used in the phylogenetic inference? Report sample sizes for all analyses (i.e. for the ILS/gene flow tests as well).

135: For clarity, specify that the "gene trees" are actually from randomly sampled sequences across the genome, not genes.

154: Explain the rationale of this BIC cutoff. Is it based on the original QuIBL paper, or used here only?

169, 224: Provide complete details of the PSMC pipeline run parameters.

172: Given the apparent signal of some ILS and gene flow, is this a robust way to estimate mutation rate? Presumably these factors could cause the mutation rate to be under-estimated. Provide a rationale or acknowledgement, and, if possible, compare with other mutation rate estimates from kiwi or similar species.

204: Are the kiwi genomes highly contiguous? How does contiguity of the assemblies potentially impact ROH identification? Add more information to the text or supplement about the contiguity of the genome assemblies.

209: Clarify if the values in Fig. 2B were obtained from ROHan with ROH length set to 1 Mb for all species.

220: What is meant by "this species"? This part is unclear.

225: Clarify that the PSMC results were checked for overfitting (did all intervals contain >=10 recombinations, as recommended by https://github.com/lh3/psmc?).

237-241: A graphical representation of the gene flow and demographic history results should be provided, if possible. It's hard to grasp these results from the text only. The authors could look to Edelman et al. 2019 for inspiration. More detailed demographic model figures are standard in the literature, and very helpful for understanding the results, but other figures to visually show the amount of gene flow/ILS could also be used.

264-271: Is the aim to look at "recent" inbreeding? How recent? Be explicit. Length of ROH corresponds with the number of generations back to the shared common ancestor. ROH <=1 Mb can originate from shared ancestors many generations ago, which may or may not be considered "recent" depending on the context, and might not even be reflective of "inbreeding" per se, but rather of limited population size. Given that the other demographic analyses deal with far more ancient history tens to hundreds of thousands of years ago, while some of the species declines occurred far more recently in the post-colonial era within the past thousand years, the communication of these results needs to be more precise.

273: It would be helpful to add the other species to this figure.

216-22 and 281-286: Why is there a "bias" for these particular genomes when aligning to a conspecific reference? Does it have to do with the same/different individual being used for the resequencing and the assembly? This is unclear. Explain.

303-308: These results are not consistent. The QuIBL analysis suggests there is little evidence of gene tree discordance overall, consistent with some ILS, but the hPSMC results suggest continuous gene flow. Later, "long-term" ILS is mentioned as a possibility (line 319), which would be surprising given the relatively limited population sizes of these species (~20k or less under the assumed mutation rate). A more coherent explanation of the results is needed in order to tie them together and support the study conclusions. I might also suggest using additional methods to address these issues, since the analyses included in this paper do not seem sufficient or convincing.

360-361, 365: Species with long generation times will not necessarily show signals of recent bottlenecks in genome-wide heterozygosity. This study does not adequately resolve whether the observed ROH are due to very recent inbreeding or more distant common ancestry. These levels of genome-wide diversity in the kiwi genomes appear largely consistent with the long-term Ne values in each species, which is not surprising. These points need to be made clear.

414-421: It is well known that PSMC provides effectively no resolution for this time period (25 kya to present). Furthermore, inspection of Fig. 3 does not appear to show a marked and consistent signal of decline among all species. This is mostly speculation and should either be rephrased to avoid misleading the reader, or removed.

435-437: The results presented in this study are insufficient to support this claim. As stated above, there is not enough power in the analyses of these single genomes to demonstrate the impact of volcanism over other forces. Furthermore, linking PSMC curves to environmental shifts is dubious, since PSMC analyses involve several over-simplifications and assumptions. For example, the results are highly dependent on the assumed mutation rate and generation time, which are often unknown (in this case, we don't know how reliable the mutation rate is).

6. PLOS authors have the option to publish the peer review history of their article (what does this mean?). If published, this will include your full peer review and any attached files.

Reviewer #1: No

Reviewer #2: No

---

## [Author Response · Author response to Decision Letter 0]

13 Jun 2022

We have attached an additional document with point by point responses to the editorial and reviewer comments

---

## [Decision Letter · Decision Letter 1]

15 Jul 2022

PONE-D-22-08093R1Genomic insights into the evolutionary relationships and demographic history of kiwiPLOS ONE

Dear Dr. Westbury,

Thank you for submitting your manuscript to PLOS ONE. After careful consideration, we feel that it has merit but does not fully meet PLOS ONE’s publication criteria as it currently stands. Therefore, we invite you to submit a revised version of the manuscript that addresses the points raised during the review process as detailed below.

We look forward to receiving your revised manuscript.

Kind regards,

Susanne P. Pfeifer

Academic Editor

PLOS ONE

Additional Editor Comments:

There remains a major concern about the reliable distinction of incomplete lineage sorting from post-divergence gene flow in this study. Currently, the authors favor one process over the other, without much convincing evidence. Additional analyses will be needed to address this issue (as well as several smaller ones highlighted by the two reviewers). Both reviewers agree that the writing itself needs to be improved as well, in particular with regards to the distinction between previously published results and novel insights from this study (which was already pointed out in the first round of reviews) and the discussion (which needs to be more focused). For reproducibility, a permanent repository for any scripts used in the analyses should be included in a future revision. 

Reviewers' comments:

Reviewer's Responses to Questions

**Comments to the Author**

1. If the authors have adequately addressed your comments raised in a previous round of review and you feel that this manuscript is now acceptable for publication, you may indicate that here to bypass the “Comments to the Author” section, enter your conflict of interest statement in the “Confidential to Editor” section, and submit your "Accept" recommendation.

Reviewer #2: (No Response)

Reviewer #3: (No Response)

2. Is the manuscript technically sound, and do the data support the conclusions?

Reviewer #2: Partly

Reviewer #3: Partly

3. Has the statistical analysis been performed appropriately and rigorously? 

Reviewer #2: No

Reviewer #3: Yes

4. Have the authors made all data underlying the findings in their manuscript fully available?

Reviewer #2: Yes

Reviewer #3: No

5. Is the manuscript presented in an intelligible fashion and written in standard English?

Reviewer #2: Yes

Reviewer #3: Yes

6. Review Comments to the Author

Reviewer #2: I have read the revised manuscript from Westbury et al. and find that although it has improved, there are outstanding issues. I am mostly satisfied that the study has been differentiated from already published works. However, my biggest concern is that the conclusions about incomplete lineage sorting (ILS) versus post-divergence gene flow are not supported by the evidence. Also, a couple of minor issues I mentioned in my original review were not adequately addressed. See below for complete details. I recommend further revision before the manuscript can be accepted for publication.

ILS/gene flow analyses: The presentation of the ILS/gene flow results are slightly more clear now, and after some further thought, I am not convinced that the results favor post-divergence gene flow over ILS. First, hPSMC simply does not distinguish between these two processes, so it is not sufficient evidence either way. Second, QuIBL is supposed to specifically differentiate between ILS+gene flow and ILS alone, and the authors simply chose to disagree with the QuIBL results tending to support the ILS alone model. The argument of site discordance within congruent windows suggesting post-divergence gene flow does not make sense to me. What seems to be happening is that kiwi diverged very recently, and may really be incipient species or even subspecies that are still capable of hybridizing. Kiwi have long generation times, apparently only diverged within the last few million years, and it is well known that bird genomes evolve slowly. Under this scenario, I would expect ILS to be more of a factor than post-divergence gene flow. In my original review, I suggested performing additional analyses to better resolve this question. I reiterate that suggestion here. One possibility might be to use the ABBA/BABA test, but there are probably other options as well.

Discussion: At about four pages, the Discussion is far too long. Compare with the results, which are just over one page in length. The Discussion needs to be condensed, and the main conclusions that are supported by the analyses in the paper should be stated clearly. As is, the Discussion is meandering, not very well organized, and hard to digest.

115-123: Briefly mention the mean or range of sequencing depths.

165: I don't advocate for using the term "locus trees" instead of "gene trees". Gene tree is the terminology used in the literature. It would be better to simply state, once, that the "gene trees" are actually just from windows, not genes.

Table S1: The previously published data (reads, assemblies) should have citations for sources. A supplemental table being too "busy" isn't really justification for failing to give proper attribution and accurately document where the data came from.

Reviewer #3: In this manuscript, the authors re-examine the evolutionary relationships between kiwi species and look for introgression between these species. I like the introduction as well as the figures and see value in publishing confirmatory results besides clear values of nucleotide diversity (for instance in Figure 2). However, as most of their results are confirmatory, which results are confirmatory and what has been gained from performing this study should be made explicitly clear. Here are my more specific comments:

1. Please provide a link (at Github or Dryad etc) to the scripts used to perform the analyses in this study.

2. In the abstract, the authors claim that they uncover similarities and differences in the demographic histories of the kiwi species studied here. However, as Weir et al (2016) already uncovers similarities in the demographic histories using PSMC, the authors should mention that they confirm previous observations of past population size changes. If the authors disagree with this, then they should mention in the abstract what they uncovered that is different.

3. Overall, I really like Figure 2. Could you perhaps also provide the synonymous site heterozygosity or heterozygosity calculated only using 4-fold degenerate sites as well? I suggest that as it may be helpful later for other studies.

4. Lines 347-360: We now know from multiple studies that unaccounted for factors such as the presence of purifying selection across the genome (Johri et al 2021) and hidden substructure in populations (Chikhi et al 2018) can result in biased inference of population history when using MSMC/ PSMC -like approaches. Although there has been no thorough investigation of how hPSMC would be biased by the same factors, it most likely would. Thus, the authors should discuss how this could affect their results.

The same reasons might be responsible for observing similar temporal patterns of population decline across all five species, as mentioned in lines 491-493.

5. This is simply a suggestion, and the authors can decide whether they like it or not. You could place the section about mapping to different reference genomes and obtaining the same species tree in the Results section (instead of the Methods section), if you like, as I believe that it is an interesting result. However, this is entirely up to the authors.

6. The discussion section currently meanders a bit. Subsections with headings will really help the reader get the main points from the discussion.

Minor Comments:

1. Lines 368-370: could you please refer to the respective figures from your manuscript.

References:

Chikhi L et al. 2018. The IICR (inverse instantaneous coalescence rate) as a summary of genomic diversity: insights into demographic inference and model choice. Heredity. 120:13–24.

Johri P et al. 2021. The impact of purifying and background selection on the inference of population history: problems and prospects. Mol. Biol. Evol. 38:2986–3003.

7. PLOS authors have the option to publish the peer review history of their article (what does this mean?). If published, this will include your full peer review and any attached files.

Reviewer #2: No

Reviewer #3: No

---

## [Author Response · Author response to Decision Letter 1]

24 Aug 2022

We have included point by point responses to the reviewers as an attachment in this submission.

---

## [Decision Letter · Decision Letter 2]

20 Sep 2022

PONE-D-22-08093R2Genomic insights into the evolutionary relationships and demographic history of kiwiPLOS ONE

Dear Dr. Westbury,

Thank you for submitting a revised version of your manuscript to PLOS ONE. Most of the reviewers' comments were addressed in this revision, however there remain a few outstanding points that should be addressed: 1) minor modifications in the text would help to improve clarity and 2) p-values should be included in Table S9 to strengthen the analysis.

We look forward to receiving your revised manuscript.

Kind regards,

Susanne P. Pfeifer

Academic Editor

PLOS ONE

Journal Requirements:

Additional Editor Comments (if provided):

Reviewers' comments:

Reviewer's Responses to Questions

**Comments to the Author**

1. If the authors have adequately addressed your comments raised in a previous round of review and you feel that this manuscript is now acceptable for publication, you may indicate that here to bypass the “Comments to the Author” section, enter your conflict of interest statement in the “Confidential to Editor” section, and submit your "Accept" recommendation.

Reviewer #2: (No Response)

Reviewer #3: (No Response)

2. Is the manuscript technically sound, and do the data support the conclusions?

Reviewer #2: Yes

Reviewer #3: (No Response)

3. Has the statistical analysis been performed appropriately and rigorously? 

Reviewer #2: Yes

Reviewer #3: (No Response)

4. Have the authors made all data underlying the findings in their manuscript fully available?

Reviewer #2: Yes

Reviewer #3: No

5. Is the manuscript presented in an intelligible fashion and written in standard English?

Reviewer #2: Yes

Reviewer #3: (No Response)

6. Review Comments to the Author

Reviewer #2: I thank the authors for addressing my previous comments. I have a few more relatively minor concerns with regard to some of the changes in the revised manuscript, as detailed below.

26-27: Would help to state which species this is referring to, or what the exceptions are.

188: Define fb

189-190: Blocks of what size were used?

307-311: These sentences are not sufficiently clear: 307: Similarities between what? the ingroup and outgroup? Is the bias in the emu's diversity, or the kiwi's? 310: Why is the bias caused by using a conspecific reference as opposed to a heterospecific reference? You don't actually know which produces the more accurate result.

310: Fewer not less

412-423: I suggest removing this part of the Discussion. It's very hard to comprehend and is better understood by visual by examination of the figures. Obviously the specifics are complicated and there are signals of gene flow/ILS throughout the phylogeny, so I recommend just summarizing with the big picture: there are signals of gene flow and/or ILS throughout the tree, which is not too surprising for a very young radiation like this. I still think the analyses of gene flow v. ILS are a little shaky, and deeper investigations of particular species histories should be left to future studies with larger datasets (ie, more individuals).

Fig 2: All cells of the matrix are a uniform shade of grey, so the results cannot be seen. Also, what is the basis for the gene flow events "hypothesized" by the authors? Explain this, or, preferably, eliminate and just show the actual results.

Table S3: Define N50, L75

Table S8: "production" should be "proportion" ?

Table S9: I have not used this method of calculating f-branch statistics, but the fact that no p-values could be computed for about half the table is very disconcerting. What is the explanation? Have the authors made an effort to rectify this? I consider this a serious flaw with this analysis.

Reviewer #3: The authors have addressed most of the comments.

The github link is not really working and so the data availability could not be verified.

7. PLOS authors have the option to publish the peer review history of their article (what does this mean?). If published, this will include your full peer review and any attached files.

Reviewer #2: No

Reviewer #3: No

---

## [Author Response · Author response to Decision Letter 2]

22 Sep 2022

I have attached the responses as a separate word document

---

## [Editor Report · Decision Letter 3]

28 Sep 2022

Genomic insights into the evolutionary relationships and demographic history of kiwi

PONE-D-22-08093R3

Dear Dr. Westbury,

We’re pleased to inform you that your manuscript has been judged scientifically suitable for publication and will be formally accepted for publication once it meets all outstanding technical requirements.

Kind regards,

Susanne P. Pfeifer

Academic Editor

PLOS ONE

---

## [Editor Report · Acceptance letter]

30 Sep 2022

PONE-D-22-08093R3 

Genomic insights into the evolutionary relationships and demographic history of kiwi 

Dear Dr. Westbury:

I'm pleased to inform you that your manuscript has been deemed suitable for publication in PLOS ONE. Congratulations! Your manuscript is now with our production department. 

Kind regards, 

on behalf of

Dr. Susanne P. Pfeifer 

Academic Editor

PLOS ONE